# Molecular Aspects of Pruritus Pathogenesis in Psoriasis

**DOI:** 10.3390/ijms22020858

**Published:** 2021-01-16

**Authors:** Kamila Jaworecka, Joanna Muda-Urban, Marian Rzepko, Adam Reich

**Affiliations:** 1Department of Dermatology, Institute of Medical Sciences, Medical College of Rzeszow University, PL-35-055 Rzeszow, Poland; kamilajaworecka@gmail.com (K.J.); joanna.muda@gmail.com (J.M.-U.); 2Institute of Physical Culture Sciences, Medical College of Rzeszow University, PL-35-055 Rzeszow, Poland; mrzepko@ur.edu.pl

**Keywords:** psoriasis, palmoplantar pustulosis, pruritus, itch, itching

## Abstract

Psoriasis is a chronic, systemic inflammatory disease with a genetic background that involves almost 3% of the general population worldwide. Approximately, 70–90% of patients with psoriasis suffer from pruritus, an unpleasant sensation that provokes a desire to scratch. Despite the enormous progress in understanding the mechanisms that cause psoriasis, the pathogenesis of psoriasis-related pruritus still remains unclear. In order to improve patients’ quality of life, development of more effective and safer antipruritic therapies is necessary. In turn to make it possible, better understanding of complexed and multifactorial pathogenesis of this symptom is needed. In this article we have systematized the current knowledge about pruritus origin in psoriasis.

## 1. Introduction

Psoriasis is a chronic, inflammatory, immune-mediated skin and joint disease with genetic background, affecting even 3% of general population. The most characteristic skin lesion of psoriasis is erythematosus plaque covered with silvery scales. Approximately 70–90% of patients with psoriasis suffer from pruritus, an unpleasant sensation that provokes a desire to scratch [1]. Due to the systemic inflammation that characterizes psoriasis, several comorbidities have been recently linked with this disease and they may also contribute to triggering, maintaining or even worsening of the psoriasis-related pruritus. The subjective and multidimensional nature of this symptom renders it challenging for clinicians and researchers to measure it appropriately and to provide optimal therapy. However, it is important to be able to control pruritus in psoriasis to prevent Koebner phenomenon (i.e., development of new psoriatic lesions due to minor trauma to apparently healthy skin) [2] and worsening of skin lesions as well as to improve patients’ quality of life. Remarkably, itching is often considered by patients as the most troublesome and unpleasant symptom of psoriasis. Pruritus in subjects with psoriasis most often appears at night and in the evening, but less frequently in the morning or around noon. According to various studies, pruritus causes difficulty in falling asleep in approximately 50–66% of patients [1,3,4,5]. Approximately 70% of patients experience itching at the sites of the lesions, in the remaining 30% it also affects unchanged skin [1,3]. More than 70% of patients experience itching on a daily basis [1,3]. The most important factors that exacerbate itching in psoriasis patients are dry skin and emotional stress, but other factors also may play a significant role [4,6]. In order to develop more effective and safer antipruritic therapies, better understanding of complexed and multifactorial pathogenesis of pruritus in psoriasis is needed. In this article we have systematized the current knowledge about pruritus origin in psoriasis.

## 2. Data Sources and Study Selection

This review was conducted using a systematic electronic literature search of the PUBMED, Mendeley and Science Direct databases. Index words included combinations of terms: “psoriasis” or “palmoplantar pustulosis” coupled with “pruritus”, “itch” or “itching”. All articles published until 1 December 2020 were taken into consideration. Our search yielded a total of 1308 results while browsing PUBMED, 2398 results containing the mentioned keywords while searching the Mendeley database and 10,808 results while browsing Science Direct. All results were checked for relevance. First, duplications and articles published in languages other than English were excluded (*n* = 9870). In the next step, non-human studies, non-clinical trials and off-topic publications were eliminated (*n* = 159). Finally, review articles, case reports, conference abstracts were excluded (*n* = 69). Ultimately, 13 research articles, focused on the pathogenesis of pruritus were included in this review (Table 1). 

## 3. Results

### 3.1. Histamine and Mast Cells

Data on histamine in psoriasis, one of the best-known pruritic mediator, remain controversial. Many investigators and experts share the opinion that histamine is not involved in pruritus associated with psoriasis. In line with this suggestion, no correlation between itch intensity and the histamine plasma level was found, and no difference in histamine plasma levels were observed between pruritic and non-pruritic patients with psoriasis [8]. Moreover, studies which measured the number of cutaneous mast cells, the major histamine producers in the human body, have shown inconsistent results while comparing pruritic vs. non-pruritic psoriasis patients [7]. In the beginning of the 21st century, Japanese scientists, as the first in the world, performed a study documenting itch-related local markers in psoriasis by assessing the number of various dermal cell types and also performing histological and immunohistochemical analysis of the skin biopsy specimens obtained from 38 patients with psoriasis vulgaris. For the purposes of the study, patients were split into two groups based on the presence or absence of pruritus. In pruritic psoriatic skin in comparison to non-itchy skin, an increased number of mast cells was observed, and these cells showed signs of increased activity [7]. Contrary to these observations, Peres et al. did not observe any significant relationship between the number of dermal mast cells and the level of itch reported by patients [16]. However, it has to be mentioned that majority of study participants were on topical and/or systemic treatment, which might have influenced the results and should be considered as a limitation of the study [16]. The use of certain psoriasis medications, such as glucocorticosteroids, cyclosporine A or acitretin, may reduce the mast cell count [20]. Interestingly, other authors, similarly to Nakamura et al., also found increased mast cell count in psoriatic skin [21]. Furthermore, Petersen et al. observed that these mast cells are hyperactivated in active psoriasis [22]. It is thus possible that histamine can be overproduced locally in the dermis and the histamine plasma level does not necessarily reflect its content in the skin. 

Despite lack of well-designed controlled studies that would confirm the effectiveness of antihistamines in psoriatic pruritus, some physicians use them to relieve itch in psoriatic patients. Authors such as Prignano et al. [23], Amatya et al. [24] or Yosipowitch et al. [25] based on their questionnaire studies, noticed some antipruritic effect of antihistamines in psoriatic patients, but each time they only paid attention to the short effectiveness of these drugs. In 2017 Domagala et al. published results of a double-blinded, randomized and placebo-controlled study evaluating the efficacy of clemastine—first generation histamine-1 receptor (H1R) antagonist, or levocetirizine—second generation H1R antagonist, in reducing pruritus in psoriasis as an addition to the standard psoriasis treatment. They found significantly higher decrease in mean visual analogue scale (VAS) scoring for the worst pruritus as well as significant reduction in the mean scoring of 12-Item Pruritus Severity Scale in clemastine and levocetirizine groups when compared to placebo. Despite favorable findings, this study had also major limitations such as a short follow-up period and the small number of observed patients (*n* = 61) [26]. However, similar results on the effectiveness of levocetirizine was described by Mueller et al. [26]. They noted that, in addition to rapid reduction in pruritus intensity, levocetirizine had also improved dermatology-related quality of life, stress, anxiety and global level of functioning [27]. 

While most attention was focused on the H1R, other histamine receptor subtypes should not be overlooked. Mommert et al. [28] found that stimulation of H4 receptor, which is highly expressed on plasmocytoid dendritic cells (pDC) in psoriasis [29], increases production of interleukin 17 (IL-17), a cytokine that plays a major role in the pathogenesis of psoriasis. Recently, it has been also shown that blockade of H4R may help to ameliorate imiquimod-induced skin inflammation, diminish epidermal hyperproliferation, and inhibit spontaneous scratching behavior in mice [30]. These observations suggest that histamine relevance in the pathophysiology of pruritus in psoriasis is still uncovered and further investigations are needed. 

### 3.2. Substance P and Other Neuropeptides

Neuropeptides are small proteins secreted from nerve endings in the central and peripheral nervous system in response to various factors such as stress, and modulate synaptic transmission [10,23]. They may activate dendritic cells, lymphocytes, macrophages and neutrophils, degranulate mastocytes, cause vascular changes in the skin, stimulate synthesis and release of many pro-inflammatory cytokines [31,32]. The imbalance of neuropeptides in psoriatic skin is being suggested to play a role in perception of itching. One of the neuropeptides, namely substance P (SP), an undecapeptide of the tachykinin family, has been implicated in the pathogenesis of pruritus for many years. Furthermore, other neuropeptides are believed to be involved in pruritus mediation. It was shown that SP, neurokinin A (NKA), and vasoactive intestinal peptide (VIP) may elicit itch upon intradermal injection into normal human skin (Table 1) [33,34].

Amatya et al. [12] investigated the response to intradermally injected SP into psoriatic skin and confirmed induction of pruritus, flare and wheal in these patients. However, there were no statistically significant differences in latency period, duration or maximum intensity of itching evoked by intradermal injected SP between psoriasis and healthy control skin [12]. Almost 20 years ago it was demonstrated that, in lesional skin from pruritic psoriasis patients, contrary to non-pruritic individuals, a significantly elevated concentration of SP was observed [7]. Moreover, investigators paid attention to a positive correlation between the number of intraepidermal SP-positive fibers in perivascular areas of lesional skin and the degree of pruritus in psoriasis [7,12]. Remröd et al. [11] and then Amatya et al. [13] assessed the expression of tachykinins (SP, neurokinin A–NKA) and their receptors (NK-1R, NK-2R) in lesional and non-lesional psoriatic skin or healthy control skin and showed an increased number of tachykinins in involved area. Furthermore, Chang et al. [9] found that expression of NK-1R was increased on keratynocytes in the psoriatic plaques of patients with pruritus. These findings indicated that drugs blocking NK-1R, such as aprepitant or serlopitant, might be an interesting treatment option in psoriatic pruritus [35,36]. In 2020, results of the phase 2 randomized, double-blind, placebo controlled clinical trial, examining the effects of serlopitant for treatment of psoriatic pruritus have been published [37]. Serlopitant reduced pruritus associated with mild to moderate psoriasis. However, the study was conducted on a small population and patients with severe psoriasis were excluded, thus further investigations are needed to confirm these observations [37]. In another study, aprepitant was shown to improve refractory chronic pruritus and quality of life in psoriatic patients [38]. Moreover, other possible pathways should be taken into consideration regarding SP. An excellent example is the transient receptor potential A1 (TRPA1), which is an ion channel that enhances SP release. Although it has never been explored in mediating psoriasis-associated itch, in a mouse model it was shown to be a necessary mediator of chronic pruritus [39]. Furthermore, TRPA1 blockade inhibited in mice behaviors associated with itching, such as scratching [40,41]. As these studies were performed on murine models of psoriasis, further investigations on this topic are warranted in human beings.

Reich et al. [10] assessed the relationships between plasma levels of selected neuropeptides, such as SP, VIP, neuropeptide Y (NPY), and calcitonin gene-related peptide (CGRP), and the presence of pruritus and its intensity in patients with psoriasis. Unexpectedly, decreased plasma levels of NPY in patients with pruritus were observed [10]. Later studies performed on mice showed that NPY signaling constitutively suppresses mechanical itch by inhibiting NPY receptor 1-expressing neurons, which are required for mechanical itch transmission in the spinal cord [41]. Based on these findings, it can be speculated that reduced NPY plasma levels in psoriasis patients aggravated mechanical itch by central modulation of this sensation. Another observation was a correlation between higher SP and VIP plasma levels with the lower pruritus intensity [10]. As SP induces itching, one may expect contrary results, i.e., positive correlation between SP levels and itch intensity [12]. However, neuropeptides may be released locally from dermal nerve endings and the reduced plasma concentrations of neuropeptides may not necessarily reflect their skin concentrations, but may result from an increased consumption or degradation of these substances in the plasma—an analogous explanation to that described for histamine. Another hypothesis takes into account the role of central nervous system (CNS) in perception of pruritus and the divergent role of neuropeptides at the periphery and in the CNS. If the level of neuropeptides in plasma corresponds to their level in the CNS rather than to the skin, then plasma and dermal levels of neuropeptides do not need to be interrelated. Nevertheless, our current understanding of the role of neuropeptides in psoriatic pruritus is far from being complete, and further investigations are needed to be able to better prevent and combat this unwanted ailment in our patients.

### 3.3. Nerve Growth Factor and Innervation

Chronic itch is associated with increased levels of nerve growth factor (NGF)—a molecule that belongs to the neurotrophic factor family [42]. This protein influences an inflammatory reaction by regulating neuropeptides, angiogenesis, cell trafficking molecules and T cell activation. Moreover, NGF exerts its action on the growth, proliferation and survival of peripheral sensory and sympathetic neurons and on a number of brain neurons [43]. Currently there two receptors for this molecule are known: high affinity tropomyosin-receptor kinase A (Trk A) and low affinity receptor p75 [44]. Nakamura et al. reported increased NGF content and increased expression of Trk A in lesional psoriatic skin with pruritus in comparison to non-pruritic skin. Additionally, the expression levels of these proteins correlated positively with the severity of pruritus [7]. Subsequent studies demonstrated that NGF expression was higher also in lesional pruritic skin than in non-lesional skin [42]. A probable consequence of elevated concentration of NGF and Trk A is elongation and branching of epidermal nerve fibers, which results in hyperinnervation. In turn, this hyperinnervation is considered to cause hypersensitivity of itch in psoriasis. However, reports of studies remain contradictory—some investigators observed increased nerve density in psoriatic skin [7], whereas others did not see such correlation [14]. This disparity may be due to different measurement techniques or heterogenous clinical history of lesions taken during biopsies. Therefore, increased nerve fiber density in the epidermis may not be an essential factor for the pathogenesis of psoriatic pruritus and further studies are needed to clarify their exact role.

### 3.4. Interleukins

The role of inflammation in psoriatic-related itch origin is undoubtedly relevant, as it is confirmed by elevated concentration of a number of inflammatory mediators and by an antipruritic effect of anti-inflammatory drugs. Various immune cells secrete cytokines that directly or indirectly may aggravate or even induce itch by increasing the inflammatory response [45]. Nakamura et al. analyzed differences in cytokine expression in the epidermis between pruritic and non-pruritic psoriatic patients. Among the tested cytokines (IFN-ɣ, TNF-α, IL-1α, IL-1β, IL-2, IL-4, IL-5, IL-6, IL-8, IL-10, IL-12) only IL-2, a SP-induced cytokine that triggers the maturation of T cells, was significantly increased in pruritic psoriatic skin [7]. Other cytokines, such as IL-4, IL-13, IL-31 and IL-33 play a key role in the pro-inflammatory and anti-inflammatory signaling pathways in patients suffering from inflammatory skin diseases such as psoriasis or atopic dermatitis [19]. In 2020, Badoor et al. published results of a study evaluating the correlation between serum concentration of IL-4, IL-13, IL-31 and IL-33 and intensity of pruritus in psoriasis and atopic dermatitis. In patients with psoriasis, similarly to atopic dermatitis, the levels of IL-4 and IL-31 were significantly elevated in comparison to healthy controls [19]. These findings were compatible with the results of another study in which elevated levels of IL-31 in the skin or serum of patients with psoriasis were demonstrated [46,47]. However, these elevated concentrations did not correlate with the intensity of itch [19]. Interestingly, Narbutt et al. proved significant reduction in both, serum IL-31 levels and severity of pruritus after narrowband ultraviolet B (UVB) phototherapy [47]. Although there are some inaccuracies in the literature, this observation might be proof that IL-31 contributes to the induction of pruritus in psoriasis. Cytokines involved in the pathogenesis of psoriasis such as IL-17, IL-22 or IL-23 are also potential agents to evoke pruritus in psoriasis, but to date data on them in relation to pruritus are limited. 

### 3.5. Vessel-Derived Molecules

Vascular abnormalities are frequently observed in psoriatic lesions and have also been suspected to be relevant in the pathogenesis of psoriasis-associated pruritus. This suggestion was supported by the positive correlation between the density of E-selectin-positive venules and the intensity of pruritus in patients with psoriasis [7]. The key role in angiogenesis of psoriatic lesions is played by the vascular endothelial growth factor (VEGF) [48]. Moreover, VEGF was also suggested to play a role in perception of pruritus in psoriasis. Higher VEGF-A expression was found in the epidermis of lesional skin from the psoriatic patients with pruritus than those without pruritus [49]. In addition, Madej et al. showed that serum concentration of vascular adhesion protein-1 (VAP-1), another adhesion molecule, was significantly higher in the group of psoriatic patients with pruritus vs. those without pruritus [50]. Prostaglandin E2 (PGE2), endothelin-1 (ET-1) and endothelial leukocyte adhesion protein 1 (ELAM-1) have also been considered to be good candidates as itch mediators in psoriasis but future studies are required to confirm this hypothesis [45].

### 3.6. Endogenous Opioids

The opioid system is considered to be a modulator of pruritus in psoriasis. It is suggested that activation of µ-opioid receptor (MOR) by a MOR ligand β-endorphin can stimulate itch, while the interaction between κ-opioid receptor (KOR) and its ligand: Dynorphin A, suppresses pruritus [51,52,53]. Opioids may also induce itch acting in the central nervous system—activation of KOR in the brain may reduce or even alleviate itch [15].

Teneda et al. followed the expression patterns of µ- and κ -opioid systems in pruritic and non-pruritic psoriatic skin as well as in healthy skin. No differences regarding µ-opioid receptor expression and β-endorphin levels in the epidermis of psoriatic patients with or without itch and healthy controls were found. However, the levels of KOR and dynorphin A were significantly decreased in the epidermis of patients with psoriasis, especially those who reported pruritus compared with the control group [14]. In an analogous study, conducted few years later in Poland, compatible results were obtained showing no significant difference in MOR system expression in both lesional and non-lesional psoriatic skin, the same as in the healthy control skin. Regarding the κ-opioid pathway, the KOR system was downregulated in the lesional pruritic psoriatic skin, and its expression was positively correlated with itch sensation [15]. These findings indicate that the imbalance in the cutaneous expression of opioid receptors and their ligands may result in disordered neuroepidermal homeostasis in psoriasis, which could potentiate the transmission of itch. Importantly, in imiquimod-induced psoriasis-like dermatitis in mice, scratching behavior was suppressed by peripheral and a central MOR antagonist or a central KOR agonist [54]. It indicates that the central opioid receptor system is also involved in the regulation of pruritus in psoriasis.

### 3.7. Lipocalin-2

Another molecule which is suspected to play an important role in the pathogenesis of pruritus in psoriasis is lipocalin-2 (LCN2). This protein, also known as 24p3 and neutrophil gelatinase-associated lipocalin (NGAL), is stored in the specific granules of human neutrophils and secreted by activated cells [55,56]. LCN2 has been associated with neurodegeneration, cancer metastasis, insulin resistance, obesity, and inflammatory responses [57,58]. Additionally, LCN2 was found to contribute to the pathogenesis of psoriasis by modulating neutrophil function to enhance T-helper 17-type responses [58]. Aizawa et al. on the group of 59 patients suffering from psoriasis observed that serum LCN2 concentration is significantly higher in this group compared to healthy controls and that plasma LCN2 level positively correlated with the intensity of pruritus [18]. These findings may indicate that LCN2 could be another mediator involved in the aggravation of pruritus in psoriasis.

### 3.8. Future Directions to Identify New Itch Mediators

Gene expression analyses is possible way to find factors involved in the pathogenesis of pruritus in psoriasis. Nattkemper et al. used RNA sequencing to analyze so called “itchscriptom” and identified several possible “itch-related” genes, including also well-known and inflammatory mediators described above, such as various cytokines (IL-17A, IL-23A, IL-31), which were commonly overexpressed in itchy atopic and psoriatic skin [17]. Nowadays, part of them is a target of biological drugs used in psoriasis therapy, e.g., IL-17A. In addition, overexpression of genes encoding SP and its receptor NK-1R in both atopic and psoriatic lesional skin was observed, a finding that further supports SP’s role in the pathogenesis of pruritus in psoriasis. In addition, elevated gene transcript levels of such genes, as phopspolipase A2 IVD and phospholipase C, voltage-gated sodium channel 1.7 (Nav1.7), transient receptor potential vanilloid 1 and 3 (TRPV1, TRPV3), transient receptor potential melastatin 8 (TRPM8) and IL-36 were also observed in itchy psoriatic skin [17]. Products of mentioned genes are potential good candidates as potential targets for new antipruritic drugs.

## 4. Conclusions

On the basis of the performed systematic electronic literature search of the PUBMED, Mendeley and Science Direct databases, several biomarkers involved in the pathogenesis of psoriasis-related pruritus were described. In summary, the molecular basis for psoriatic pruritus is a result of complex interaction between the nervous, neuroendocrine, immune and vascular system and is still not fully understood. As a consequence, there is a lack of effective antipruritic treatment for psoriatic patients. Thus, further studies are urgently needed to provide clarification of the mechanisms involved in the pruritus in psoriasis to develop better medications for psoriatic itch. Medical history plays a pivotal role in determining the pruritus causes also in psoriatic patients and should drive the anti-pruritic therapy. Since pruritus is a significant determinant of patients’ quality of life, physicians should be aware that even effective anti-psoriatic therapies may not necessarily control pruritus or the occurrence of pruritus in responders to anti-psoriatic treatment may be a fist symptom of growing secondary unresponsiveness. 

## Figures and Tables

**Table 1 ijms-22-00858-t001:** Studies on mechanisms of pruritus in psoriasis (↓—decreased, ↑—increased, ↔—the same, BDNF—brain-derived neurotrophic factor, CGRP—calcitonin-gene related peptide, ELAM-1—endothelial leukocyte adhesion molecule 1, ICAM-1—intercellular adhesion molecule 1, INF—interferon, IL—interleukin, KOR—κ-opioid receptor, LCN2—lipocalin 2, MOR—µ-opioid receptor, NEP—neutral endopeptidase, NGF—nerve growth factor, NK-1R–neurokinin-1 receptor, NK-2R—neurokinin-2 receptor, NKA—neurokinin A, NPY—neuropeptide Y, NT-3—neurotrophin 3, OPRK1—opioid receptor kappa 1, OPRM1—opioid receptor mu 1, PECAM-1—platelet endothelial cell adhesion molecule-1, p75NTR—p75 neurotrophin receptor, PACAP—pituitary adenylate cyclase-activating peptide, PGP 9.5—protein gene product 9.5, PMN—polymorphonuclear leukocytes, Sema3A—semaphorin 3A, SOM—somatostatin, SP—substance P, TNF-α—tumor necrosis factor α, TRPM8—transient receptor potential melastatin 8, TRPV—transient receptor potential vanilloid, Trk A—tropomyosin-related kinase A, VCAM-1—vascular cell adhesion molecule 1, VIP–vasoactive intestinal peptide, VPACR–vasoactive intestinal peptide receptor).

Study	Number of Included Patients	What Was Evaluated?	Major Findings
Nakamura M et al., 2003 [7]	Psoriasis: *n* = 38 (23 with pruritus and 15 without pruritus)	Number of mast cells, Langerhans cells, macrophages, as well as expression of PGP 9.5, SP, CGRP, VIP, SOM, NPY, NGF, NGF-receptor (Trk A), BDNF, NT-3, NEP, angiotensin-converting enzyme, INF-ɣ, TNF-α, IL-1α, IL-1β, IL-2, IL-4, IL-5, IL-6, IL-8, IL-10, IL-12, PMN, PECAM-1, ICAM-1, ELAM-1, VCAM-1 in pruritic vs. non-pruritic psoriatic skin	↑ mast cells in the dermis of pruritic vs. non-pruritic psoriatic skin ↑ NGF-immunoreactive keratynocytes, ↑ expression of Trk A in the epidermis and dermal nerve fibres, ↑ PGP 9.5-immunoreactive nerve fibers in the epidermis and in the upper dermal areas, ↑ SP-containing nerves in the perivascular areas of pruritic in comparison to non-pruritic psoriatic skin ↑ IL-2-immunoreactive cells in pruritic vs. non-pruritic psoriatic skin ↑ ELAM-1-positive venules in pruritic compared to non-pruritic psoriatic skin ↓ expression of NEP in the epidermal basal layer and in the endotelia of blood vessels in pruritic vs. non-pruritic samples ↔ expression of CGRP, SOM ↔ INF-ɣ, TNF-α, IL-1α, IL-1β, IL-4, IL-5, IL-6, IL-8, IL-10, IL-12 expression in the epidermis and infiltrating mononuclear cells ↔ ICAM-1 and VCAM-1-immunoreactive vessels in the upper dermis and ICAM-1-positive vessels in the epidermis of pruritic and non-pruritic psoriatic skin
Wisnicka B et al., 2004 [8]	Psoriasis: *n* = 59 (43 with pruritus and 16 without pruritus) Healthy controls: *n* = 32	Plasma level of histamine, SP, CGRP	↑ CGRP plasma levels in pruritic psoriatic patients vs. healthy controls ↔ histamine and SP plasma concentration in all the group No correlations between CGRP, histamine or SP levels and pruritus intensity
Chang SE et al., 2007 [9]	Psoriasis: *n* = 20 (10 with pruritus and 10 without pruritus) Healthy controls: *n* = 10	Expression of NGF, TrkA, p75NTR, NT4, CGRP, CGRP receptor, SP, NK-1R, VIP, PACAP, VPACR, NEP, PGP 9.5, collagen VII in lesional pruritic psoriatic skin vs. non-pruritic psoriatic skin, non-lesional psoriatic pruritic skin and healthy skin	↑ expression of SP receptors, TrkA and CGRP receptors in keratynocytes and number of dermal nerves in pruritic compared with non-pruritic lesional psoriatic skin ↔ expression of assessed neuropeptides and NEP between the pruritus and non-pruritus groups
Reich A et al., 2007 [10]	Psoriasis: *n* = 59 (43 with pruritus, 16 without pruritus) Healthy controls: *n* = 32	Plasma concentration of SP, CGRP, VIP and NPY	↓ NPY plasma levels in patients with pruritus vs. without pruritus ↔ SP, CGRP and VIP plasma concentration in pruritic and non-pruritic psoriatic patients Negative correlation between pruritus intensity and SP or VIP plasma levels No correlation between pruritus intensity and CGRP or NPY plasma levels
Remröd C et al., 2007 [11]	Psoriasis: *n* = 13	Expression of SP and the NK-1R in involved and noninvolved psoriatic skin	No correlation between SP-positive nerve fibers nor SP-positive cells and the level of pruritus
Amatya B et al., 2010 [12]	Psoriasis: *n*=15 (with pruritus) Healthy controls: *n* = 15	Pruritus, flare and wheal after injection of SP, saline and histamine	SP induced pruritus, flare and wheal in both psoriasis patients and healthy controls (no significant differences between studied groups)
Amatya B et al., 2011 [13]	Psoriasis: *n* = 28 Healthy controls: *n* = 10	Expression of SP, NKA, NK-1R and NK-2R in lesional, non-lesional and healthy skin	Positive correlation between the pruritus intensity and the number of SP-positive nerve fibers and number of NK-2R-immunoreactive cells in the lesional skin
Taneda K et al., 2011. [14]	Psoriasis: *n* = 24 Healthy controls: *n* = 5	Number of epidermal nerve fibers, the levels of Sema3A and the expression patterns of µ- and κ-opioid systems in pruritic and non-pruritic psoriatic skin and healthy skin	↔ expression of µ-opioid receptor and expression levels of β-endorphin in the epidermis of all analyzed groups ↓ expression of κ -opioid receptor in psoriatic pruritic skin compared to healthy skin ↓ dynorphin-A levels in the epidermis of pruritic psoriatic patients compared with healthy controls
Kupczyk P et al., 2017. [15]	Psoriasis: *n* = 20 Healthy controls: *n* = 20	Opioid receptor genes (OPRM1, OPRK1) and protein (MOR, KOR) expression in lesional and non-lesional psoriatic skin and healthy control	↓ expression of KOR in lesional psoriatic skin with itch in comparison with lesional skin without itch ↔ OPRK1 expression in groups with and without pruritus ↔ expression of OPRM1/MOR system in groups with or without pruritusNegative correlation between OPRK1/KOR pathway and intensity of pruritus. No correlation between the OPRM1/MOR expression and severity of pruritus
Peres LP et al., 2018 [16]	Psoriasis: *n* = 29	Number of mast cells in the dermis of lesional-skin	No correlation between the intensity of pruritus and mast cell count
Nattkemper LA et al., 2018 [17]	Psoriasis: *n* = 25 Atopic dermatitis: *n* = 25 Healthy controls: *n* = 39	Genetic expression profiles	↑ expression of genes for IL-17A, IL-23A, and IL-31, phospholipase A2 IVD, SP, voltage-gated sodium channel 1.7, TRPV1, TRPM8, TRPV3, phospholipase C, IL-36α/γ in pruritic psoriatic skin vs. healthy controls Overexpression of phospholipase A2 IVD, SP, voltage-gated sodium channel 1.7 and TRPV1 genes in itchy skin were positively correlated with pruritus intensity
Aizawa *n* et al., 2019 [18]	Psoriasis: *n* = 59 Atopic dermatitis: *n* = 47 Healthy controls: *n* = 47	LCN2 serum concentrations	↑ LCN2 plasma concentration correlated positively with itch intensity in psoriatic patients
Bodoor K et al., 2020. [19]	Psoriasis: *n* = 59 Atopic dermatitis: *n* = 56 Healthy controls: *n* = 49	Serum levels of IL-4, IL-13, IL-31, IL-33	The levels of measured interleukins in psoriasis did not correlate with itch severity

## Data Availability

Data availability is not applicable to this article as no data sets were generated or analyzed during the current study.

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
