# Peer review of "Molecular Aspects of Pruritus Pathogenesis in Psoriasis"

_ijms, 2021, doi:10.3390/ijms22020858_

Round 1
Reviewer 1 Report
The manuscript by Jaworecka et al. is a very comprehensive review of the possible molecular mediators involved in one of the symptoms of psoriasis, pruritus. The literature search has been conducted with enough rigor. Besides the language used is very correct and with very few typographical errors. However, there is a major concern, two minor comments or suggestions and two minor typographical errors:
- Major concern: the authors claim this is a “systemic” electronic literature search. First, the word “systemic” is wrongly used (lines 32 and 281), it means “relating to a system, especially as opposed to a particular part”. Second, caution should be taken to define this review as “systematic”, since the methodology used in such studies is quite specific, a step-by-step protocol must be executed, and certain criteria must be met. The following links maybe helpful:
https://training.cochrane.org/handbook/current
https://worldwidescience.org/topicpages/e/electronic+literature+searches.html
A simple suggestion, the authors should just mention that the review was conducted using systematic literature review methodology, which is true, using a proper citation; but should refrain of characterize this study as a systematic review.
- Minor comments:
- In the result sections all the subheadings are related to mediators, except two: 5. Vascular abnormalities and 3.8. Gene expression. This is in striking contrast with the rest of the subheadings and should be corrected. In fact, in the last one (3.8) some of the mediators had previously been described in other parts.
- Table 1: The authors should clarify the criteria used to order the studies presented. The most common is in chronological order.
3. Minor typographical errors:
Line 214, the “an” before “another” seems not necessary.
Line 268, the quotation mark is not correct: ,, should be replaced by “
Author Response
We are grateful to the reviewer for his valuable comments. Following modifications have been made to the original manuscript:
Reviewer: The authors claim this is a “systemic” electronic literature search. First, the word “systemic” is wrongly used (lines 32 and 281), it means “relating to a system, especially as opposed to a particular part”. Second, caution should be taken to define this review as “systematic”, since the methodology used in such studies is quite specific, a step-by-step protocol must be executed, and certain criteria must be met. The following links maybe helpful:
https://training.cochrane.org/handbook/current
https://worldwidescience.org/topicpages/e/electronic+literature+searches.html
A simple suggestion, the authors should just mention that the review was conducted using systematic literature review methodology, which is true, using a proper citation; but should refrain of characterize this study as a systematic review.
Aurthors: We fully agree with the reviewer. The term "systemic" has been changed to "systematic". We still have the opinion, that we performed a systematic literature review to identify relevant papers, but this review cannot be termed as a systematic review, as indicated by the reviewer.
Reviewer: In the result sections all the subheadings are related to mediators, except two: 5. Vascular abnormalities and 3.8. Gene expression. This is in striking contrast with the rest of the subheadings and should be corrected. In fact, in the last one (3.8) some of the mediators had previously been described in other parts.
Authors: The subheadings have been modified to maintain the uniform format of the entire manuscript. The subchapter on "Gene expression" has been slightly modified to underline, that the new molecular techniques enable to identify new, yet not described itch mediators. We do hope, that the reviewer will be satisfied with such modifications.
Reviewer: Table 1: The authors should clarify the criteria used to order the studies presented. The most common is in chronological order.
Authors: As suggested, the studies mentioned in the table were presented in chronological order.
Reviewer: Minor typographical errors: Line 214, the “an” before “another” seems not necessary. Line 268, the quotation mark is not correct: ,, should be replaced by “
Authors: Both typographical errors have been corrected.
Reviewer 2 Report
I read with great interest the manuscript by Kamila Jaworecka et colleagues titled "Molecular aspects of pruritus pathogenesis in psoriasis" that summarize the current knowledge and understanding of psoriasis-related pruritus.
I have only minor criticisms to extend to the authors to improve the overall quality of the manuscript.
ABSTRACT
Please change this sentence "
Psoriasis is a chronic, inflammatory, immune-mediated skin and joint disease with genetic background, affecting even 3% of general population" as follow"Psoriasis is a chronic, systemic inflammatory diseases with a genetic background that involves almost 3% of the general population worldwide.
Please in the second sentence quote directly the word "itch" or "pruritus"
INTRODUCTION
Please add this paragraph to let readers understand the multi-dimensionality of pruritus in psoriasis: "Due to the systemic inflammation that characterize psoriasis, several comorbidities were recently linked and they may contribute to trigger, maintain or even worse the psoriasis-related pruritus.
Please explain add a reference in the pruritus and Koebner: Sanchez DP, Sonthalia S. Koebner Phenomenon. 2020 Dec 1. In: StatPearls [Internet]. Treasure Island (FL): StatPearls Publishing; 2020 Jan–. PMID: 31971748.
Please describe more the the pruritus characteristics [10.1111/jdv.15539,10.1111/exd.14071] and its circadian characteristics/modulation [10.1111/bjd.15469, 10.1080/07420528.2019.1678629]
DATA SOURCES
Please add the time frame per each databank you employed
3.2. Substance P and other neuropeptides
In this paragraph authors did not mention a recent article that represent a clinical proof of concept on the neurokinins antagonism as promising anti-pruritic therapy [10.1080/09546634.2020.1840502].
Please in the conclusion authors should stress also these two concepts:
- medical history plays a pivotal role in establish the pruritus causes also in psoriatic patients and should drive the anti-pruritic therapy.
- since pruritus is a heavy determinant of patients' quality of life, physicians should be aware that also effective anti-psoriatic therapies may last pruritus or even the occurrence of pruritus in patients responders may be a symptom of growing unresponsiveness.
Author Response
We are grateful to the reviewer for his time spent on our manuscript and for all supportive comments and suggestions to improve our manuscript. Following modification have been made:
Reviewer: I read with great interest the manuscript by Kamila Jaworecka et colleagues titled "Molecular aspects of pruritus pathogenesis in psoriasis" that summarize the current knowledge and understanding of psoriasis-related pruritus. I have only minor criticisms to extend to the authors to improve the overall quality of the manuscript.
Authors: We would like to thank you for such a supportive opinion about our manuscript.
Reviewer: Please change this sentence: Psoriasis is a chronic, inflammatory, immune-mediated skin and joint disease with genetic background, affecting even 3% of general population" as follow"Psoriasis is a chronic, systemic inflammatory disease with a genetic background that involves almost 3% of the general population worldwide. "Please in the second sentence quote directly the word "itch" or "pruritus"
Authors: The abstract has been modified as suggested by the reviewer.
Reviewer: Please add this paragraph to let readers understand the multi-dimensionality of pruritus in psoriasis: "Due to the systemic inflammation that characterize psoriasis, several comorbidities were recently linked and they may contribute to trigger, maintain or even worse the psoriasis-related pruritus."
Authors: The above statement has been added to the Introduction.
Reviewer: Please explain add a reference in the pruritus and Koebner: Sanchez DP, Sonthalia S. Koebner Phenomenon. 2020 Dec 1. In: StatPearls [Internet]. Treasure Island (FL): StatPearls Publishing; 2020 Jan–. PMID: 31971748.
Authors: As suggested, we have included the reference and provide a statement regarding the relationship between itching and Kebner phenomenon in maintaining psoriatic lesions.
Reviewer: Please describe more the the pruritus characteristics [10.1111/jdv.15539,10.1111/exd.14071] and its circadian characteristics/modulation [10.1111/bjd.15469, 10.1080/07420528.2019.1678629]
Authors: In the Introduction, we have added some more information regarding pruritus characteristics including also suggested references.
Reviewer: DATA SOURCES: Please add the time frame per each databank you employed
Authors: The time frame regarding the literature search has been added.
Reviewer: Substance P and other neuropeptides: In this paragraph, authors did not mention a recent article that represents a clinical proof of concept on the neurokinin antagonism as promising anti-pruritic therapy [10.1080/09546634.2020.1840502].
Authors: In the revised manuscript the respective study has been cited.
Reviewer: Please in the conclusion authors should stress also these two concepts:
- medical history plays a pivotal role in establish the pruritus causes also in psoriatic patients and should drive the anti-pruritic therapy.
- since pruritus is a heavy determinant of patients' quality of life, physicians should be aware that also effective anti-psoriatic therapies may last pruritus or even the occurrence of pruritus in patients responders may be a symptom of growing unresponsiveness.
Authors: The conclusions have been modified accordingly.
Reviewer 3 Report
Very valuable and comprehensive review paper gathering most actual data regarding complexity of psoriasis-related pruritus. The authors throrughly performed electronic literature research and highlighted the need for development and including anti-pruritic treatment in antipsoriatic management.
Author Response
We are very grateful to the reviewer for a very supportive opinion regarding our manuscript. We do hope, that the manuscript will be of interest to the readers of the IJMS